# Agricultural Bioinputs Obtained by Solid-State Fermentation: From Production in Biorefineries to Sustainable Agriculture

Thiago Moura Rocha *[ID], Paulo Ricardo Franco Marcelino *, Rogger Alessandro Mata Da Costa [ID], Daylin Rubio-Ribeaux [ID], Fernanda Gonçalves Barbosa and Silvio Silvério da Silva [ID]

Department of Biotechnology, School of Engineering of Lorena, University of São Paulo, Lorena 12602-810, Brazil; rogger.alessandro@hotmail.com (R.A.M.D.C.); drubioribeaux@yahoo.es (D.R.-R.); fernanda.g.barbosa@usp.br (F.G.B.); silviosilverio@usp.br (S.S.d.S.)
* Correspondence: thi.rocha1995@gmail.com (T.M.R.); paulorfm@usp.br (P.R.F.M.)

**Abstract:** Agriculture plays a major role on society, especially in developing countries which rely on commodity exportation markets. To maintain high crop productivity, the use of agrochemicals was once employed as the main strategy, which in turn affected soil, water, and human health. In order to aid this issue, identifying some alternatives, such as the implementation of biofertilizers and inoculants as bioinputs in modern agriculture, are imperative to improve ecosystem quality. Among these bioinputs, a few bioproducts have shown good performances, such as phytohormones (e.g., auxins and giberellins), biosurfactants, and other enzymes; thus, it is extremely important to assure the quality and feasibility of their production in biorefinery scenarios. These bioproducts can be synthesized through fermentation processes through utilizing plant biomasses and agricultural byproducts as carbon sources. In this sense, to increase the tecno-economical availability of these processes, the implementation of solid-state fermentation (SSF) has shown great potential due to its ease of operation and cost-attractiveness. Therefore, this study aims to describe the main substrates used in SSF systems for the production of potential bioinputs; their associated operation hurdles, parameters, and conditions selection; the most suitable microorganisms; and the underlying mechanisms of these molecules in soil dynamics. Within this context, this study is expected to contribute to the development of new processes in modern biorefineries and to the mitigation of environmental impacts.

**Keywords:** agricultural bioinputs; plant biomass; biorefineries; solid-state fermentation

## 1. Introduction

Since the Upper Paleolithic, agriculture has evolved from primitive, almost "artisanal", methods focused on individual or small-group feeding to mechanized methods involving the intensive use of synthetic inputs and even the adoption of genetic engineering in cultivars [1,2]. Currently, agriculture is the economic basis of several countries, especially developing ones, which comprise the main exporters of agricultural products. On the other hand, agriculture in the Contemporary Era, especially after the Green Revolution that began in the 1960s in the USA and Europe, has started to focus on productivity, income, and profit rather than subsistence, as was the case in the past [3].

Current agriculture has become more harmful than symbiotically helpful, establishing a destructive relationship with land in an attempt to increase productivity and profit [4]. In recent years, the adoption of synthetic agrochemical processes and products has been observed, and these synthetic processes and products contaminate soil, water sources, and the air, causing environmental imbalances ranging from pest invasions due to insects and microorganisms to climate change [5].

In recent years, sustainable agriculture has been presented as an alternative to avoid the problems previously reported, making the agricultural techniques and products used

less harmful to the planet. The use of agricultural bioinputs has been well evaluated and increasingly explored among small- and medium-sized rural producers in several developed countries [6]. The global market size for bioinputs was estimated to be worth USD 8.8 billion in 2019 and is expected to grow at a CAGR of 13.6% to reach a value of USD 18.9 billion by 2025 [7].

Bioinputs are products based on microbial cells and metabolites, plant materials, organic or natural materials, and they are used in agriculture to combat pests and diseases, improving soil fertility and nutrient availability for plants. Furthermore, the foods produced from agricultural practices that use bioinputs are free from the agrochemical residues that are associated with the risk of neurodegenerative diseases and cancers. When compared to synthetic products, bioinputs have low toxicity and high biodegradability, which reduces environmental and human health impacts. Geopolitical and economic factors are also preponderant for the adoption of bioinputs in agriculture, since several commodity-producing countries import synthetic fertilizers that are essential for agricultural activity and food security from Canada and Russia, thus impacting trade balances. The most common examples of bioinputs that can be found on the market today are agricultural inoculants, biofertilizers, biosurfactants, phytohormones, and biological control agents. These bioproducts are obtained mainly by fermentative processes, generally by solid-state fermentation (SSF) [7,8].

SSF is an ancient technique that was first used in 2600 BC in Egypt before passing through Asia, Africa, and Europe. The technique was used in the production of some traditional foods and drinks, and from the mid-19th century onwards, it became popular in the Western world, being used in the production of enzymes, medicines, organic acids, foods, and products for agricultural use [9]. In this type of bioprocess, solid substrates, generally of natural origin, are used as a support or nutritional substrate in an environment with minimal water activity, allowing the microorganisms to grow and driving its metabolism to the target of interest [10]. As previously reported, several bioinputs used in agriculture are currently produced through SSF. The present literature review aims to describe the ways in which some bioinputs of agricultural interest are produced via SSF, the advantages and disadvantages of these bioprocesses, and the main challenges to improving these processes and increasing yields in order to reach a larger market and promote the development of sustainable agriculture.

## 2. Production of Agricultural Bioinputs by Solid-State Fermentation: General Concepts

Agricultural bioinputs are products based on microbial cells and metabolites, plant materials, or even animal materials that are used in agricultural production to combat insects and phytopathogenic microorganisms, improving soil fertility and maintaining plant fitness. Among the existing bioinputs, biofertilizers, inoculants, and biocontrol agents stand out. These biological compounds present numerous environmental advantages for farmers and consumers. The main advantages include greater sustainability with respect to the production process; reduced application costs; the greater biological fixation of nitrogen, phosphorus, and potassium micronutrients; and an increased production of substances that act on plant growth, immune modulators, and antimicrobials [11].

Different microorganisms can be used in the production of bioinputs, such as bacteria, filamentous fungi, and yeasts. Generally, the microorganisms are isolated from the plants themselves and can be classified as rhizospheric, epiphytic, or endophytic. Knowledge of the part of the plant from which the microorganism is isolated is important. Indeed, a microorganisms' application in an inoculant formulation will highly depend on its origin, and inoculant formulations can improve the lifespan and proliferation of plants. The methods of application range from seed coating application, root application, and foliar application, as well as incorporation directly into the soil to improve nutrient availability. Examples of microorganisms that promote plant growth include bacteria of the genera *Rhizobium*, *Azospirillum*, *Bacillus*, *Streptomyces*, *Gluconoacetobacter*, and *Pseudomonas* and filamentous mycorrhizal fungi such as *Claroideoglomus*, *Glomus*, and *Rhizophagus*, which are

used in the production of bioinoculants [12–14]. Recently, there has also been an increase in studies on yeasts such as those of the genera *Candida* spp., *Rhodotorula* spp., *Cryptococcus* spp., and *Saccharomyces* sp., which have been reported to promote plant growth [15].

With the aim of outlining lower-emission bioprocesses, combined with the sustainable and eco-friendlier nature of bioinputs compared to synthetic agrochemicals, the current focus of several studies is to realize the production of different plant growth promoters by utilizing renewable and low-cost feedstock in fermentation processes [16]. In this sense, solid-state fermentation (SSF) stands out for its versatility in terms of the use of raw materials. In fact, agricultural byproducts such as lignocellulosic biomasses, brans, and others can be used in SSF to produce a variety of bio-based products, including bioinputs [17–19]. SSF is commonly recognized as a process in which microorganisms are cultivated on insoluble solid substrates or in a solid matrix embedded with a limited availability of moisture (low water activity), allowing for microbial growth [20,21].

Despite the fact that a few drawbacks must still be overcome, in comparison to other techniques such as submerged fermentation (SmF), the conduction of fermentation processes under SSF exhibits attractive advantages, as summarized in Table 1. As highlighted by Manan and Webb [22], SSF offers several advantages in terms of techno-economic, environmental, and biological characteristics. However, when comparing SSF with submerged fermentation (SmF), it can be observed that SmF provides better control of environmental parameters, a broader spectrum of applications, and system homogeneity. Furthermore, these features ensure predictable and reliable product uniformity through different kinetic models and operation mode strategies [23–26]. Despite these disadvantages, SSF is still preferable for the production of agricultural bioinputs in on-farm mode due to its ease of installation, ease of maintenance for the farmer, and the lower costs associated with it.

**Table 1.** Advantages and disadvantages of using SSF as a fermentation strategy for the production of bioproducts.

| Advantages | Disadvantages |
| --- | --- |
| • Compositional complexity of the substrate used to prepare the cultivation medium to meet microbial nutritional needs; | • Decreased diversity of microbes that grow in restricted water conditions; |
| • Contamination control and absence of foam; | • Limitations in controlling humidity, temperature, and pH; |
| • Better oxygenation; | • Lower effective mixing; |
| • Savings on energy costs and in the preparation of cultivation media; | • Low-purity bioproducts are obtained, requiring greater expenditure on purification processes; |
| • Less water used in the up-stream stages, which results in lower expenses; | • The pretreatment used to prepare the substrate can be expensive; |
| • Greater productivity and income; | • Difficulty in scaling and automating processes. |
| • Decreased production of liquid waste that needs to be treated; | |
| • More concentrated final products. | |

Based on Manan and Webb [22].

In SSF, the use of agricultural byproducts serves as a solid matrix (inert support or support) and/or substrate (nutrient source) for microbial fermentation, allowing for the generation of various biological-based products. The choice of which substrate to use is extremely dependent on microbial metabolic requirements such as the carbon and nitrogen balance and the balance of other micronutrients. The structure, chemical composition, and porosity of a substrate also influence microbial development; thus, an appropriate choice of material is fundamental to facilitate its growth. Due to these aforementioned properties, a few byproducts must be pretreated (e.g., lignocellulosic biomasses) in order

to promote the fragmentation of recalcitrant components, the disruption of the fibers, and detoxification, hence facilitating enzymatic access to the substrate. Table 2 displays a variety of pretreatment techniques that are employed in current agricultural practice, including chemical, physical, physical–chemical, and biological techniques, and mainly used to treat lignocellulosic materials.

**Table 2.** Types of pretreatments employed to treat lignocellulosic biomasses for use in fermentation processes.

| Pretreatment | Methods | Advantages | Disadvantages |
|---|---|---|---|
| Physical | • Milling;<br>• Mechanical extrusion;<br>• Microwave;<br>• Ultrasound;<br>• Pyrolysis;<br>• Pulsed electric field. | • Reduction in the degree of crystallinity and degree of polymerization of cellulose;<br>• Increased surface area. | • High energy consumption;<br>• Does not remove lignin. |
| Chemical | • Dilute acid;<br>• Mild álcali;<br>• Ozonolysis;<br>• Organosolv;<br>• Ionic liquids;<br>• Deep eutectic solvents;<br>• Natural deep eutectic solvents. | • High glucose yield;<br>• Partial or total solubilization of lignin (delignification);<br>• Decrease in cellulose crystallinity;<br>• Increase in the surface area of biomass;<br>• Decrease in the degree of polymerization of cellulose. | • Acid and alkaline treatments can promote the corrosion of equipment;<br>• The high costs of some reagents and materials used to manufacture the equipment;<br>• Difficulty in recovering reagents;<br>• The formation of hydrolysis and fermentation inhibitory byproducts. |
| Physico-chemical | • Steam explosion;<br>• Hydrothermal (liquid hot water);<br>• SPORL;<br>• Ammonia-based;<br>• CO2 explosion;<br>• Oxidative pretreatment;<br>• Wet oxidation;<br>• Hydrodynamic cavitation. | • Partial or total solubilization of hemicelluloses;<br>• High yields of glucose and hemicellulose sugars. | • Formation of degradation products;<br>• It may or may not require delignification after pretreatment. |
| Biological | • Enzymatic or microbial | • Removal of a significant amount of lignin;<br>• No formation of fermentation inhibitory products. | • Requires longer retention times;<br>• Requires monitoring during the growth of microorganisms;<br>• The consumption of carbohydrates from biomass and the consequent reduction in sugar yield. |

Based on Gueri et al. [27].

The pretreatments used in lignocellulosic biomass aim to promote its fragmentation, increase its surface area and solubility, lower the cellulose crystallinity and lignin content, and increase the porosity of the cellulose fiber to facilitate microbial growth [28]. Note that the choice of the pretreatment process must take into account parameters such as efficiency, waste generation, energy requirements, and cost-attractiveness.

In SSF, in addition to the selection of supports/substrates, variables such as the standardization of particle size (granulometry), quantity and age of inoculum, specific additives, pH, and process temperature must also be considered [22]. Another parameter of fundamental importance for the performance of the SSF is the selection of the bioreactor, including its geometry and configuration, which influence the mass and energy transfer of the system. The main types of bioreactors for SSF are tray bioreactors, packed beds, air-solid fluidized bed stirred bioreactors, and rotating drum bioreactors. Tray fermenters are the most commonly used bioreactors for SSF due to their simple design [29].

Given the world's current socioeconomic and environmental scenario, there is an urgent need to adopt sustainable processes and products aimed at developing bioeconomy. Therefore, several bioproducts can be synthesized using lignocellulosic biomasses in biore-

fineries, meeting the requirements for more ecological practices. In the following sections, some important agricultural bioinputs, as well as their characteristics and production parameters in the context of SSF, are presented.

## 3. Inoculants and Biological Controlling Agents as Bioinputs

The advancement of green and circular economy concepts worldwide has intensified the search for sustainable and highly efficient products that can replace synthetic products derived from petroleum and mineral extraction. Recently, in the agricultural sector, the excessive use of NPK fertilizers (nitrogen, phosphate, and potassium compounds) and synthetic pesticides has shown the harmful effects of these toxic compounds in the reduction of the biodiversity present in soils, as well as the contamination of water sources, groundwater, and food, causing several health problems for organisms living on the planet [30,31]. An alternative to replace these synthetic products and restore the balance of soil biodiversity is the use of inoculants and biological control agents [32].

Inoculants are biological products composed of living microorganisms that benefit the development of different plant species, increasing the availability of nutrients for plants and improving the absorption of nutrients by plants. They also induce plant immunity and favor soil quality [33,34]. The microorganisms used in the preparation of inoculant formulations are known as plant growth promoters, examples of which include bacteria and fungi which can be found adhered to the roots (rhizospheric), inside plant tissues (endophytic), or in plant leaves. These microorganisms stand out for being nitrogen fixers (diazotrophs); phosphate solubilizers; and producers of siderophores, phytohormones, antimicrobial compounds, and enzymes. Examples of microorganisms used in inoculant formulations include bacteria of the genera *Burkholderia*, *Pantoea*, *Enterobacter*, *Pseudomonas*, *Massilia*, *Sphingobium*, *Sphingomonas*, *Agrobacterium*, *Rhizobium*, *Bradyrhizobium*, and *Ochrobactrum* and fungi of the genera *Penicillium* and *Mycorrhiza* [12,35,36]. A schematic representation of the interactions among those microorganisms occurring in the rhizosphere is exhibited in Figure 1.

Biological control agents are bioproducts composed of micro or macroorganisms that act to reduce the biotic and abiotic factors that may negatively interfere in agricultural production [37]. They are used in important methods for protecting plants against attacks by arthropod pests and phytopathogenic microorganisms, proving to be effective without inflicting ecological damage [38]. This study will focus on the production of microbial biological control agents or bioinsecticides, more specifically those composed of entomopathogenic bacteria and fungi, which infect insect pests and have antagonistic activity against phytopathogenic microorganisms.

Among the most common formulations of these biological control agents on the market, those containing bacteria from the genera *Agrobacterium*, *Bacillus*, *Pseudomonas*, *Streptomyces*, and *Paenibacillus* and also fungi from the genera *Trichoderma*, *Metarhizium*, and *Beauveria* are some examples of effective and well-studied agents that act against several plant diseases [12,16]. Biological control agents may act under hyperparasitism, competition, the secretion of lytic enzymes, and plant growth promotion, inducing systemic host resistance and antibiosis [39].

In recent years, there has been an increasing number of works regarding the production and effects of inoculants and biological control agents, as depicted in Figure 2.

From a technological point of view, the increasing study of inoculants and biological control agents is due to the advancements in fermentative processes and molecular biology techniques, the development of biorefinery concepts, and the imperative need for more sustainable agricultural inputs.

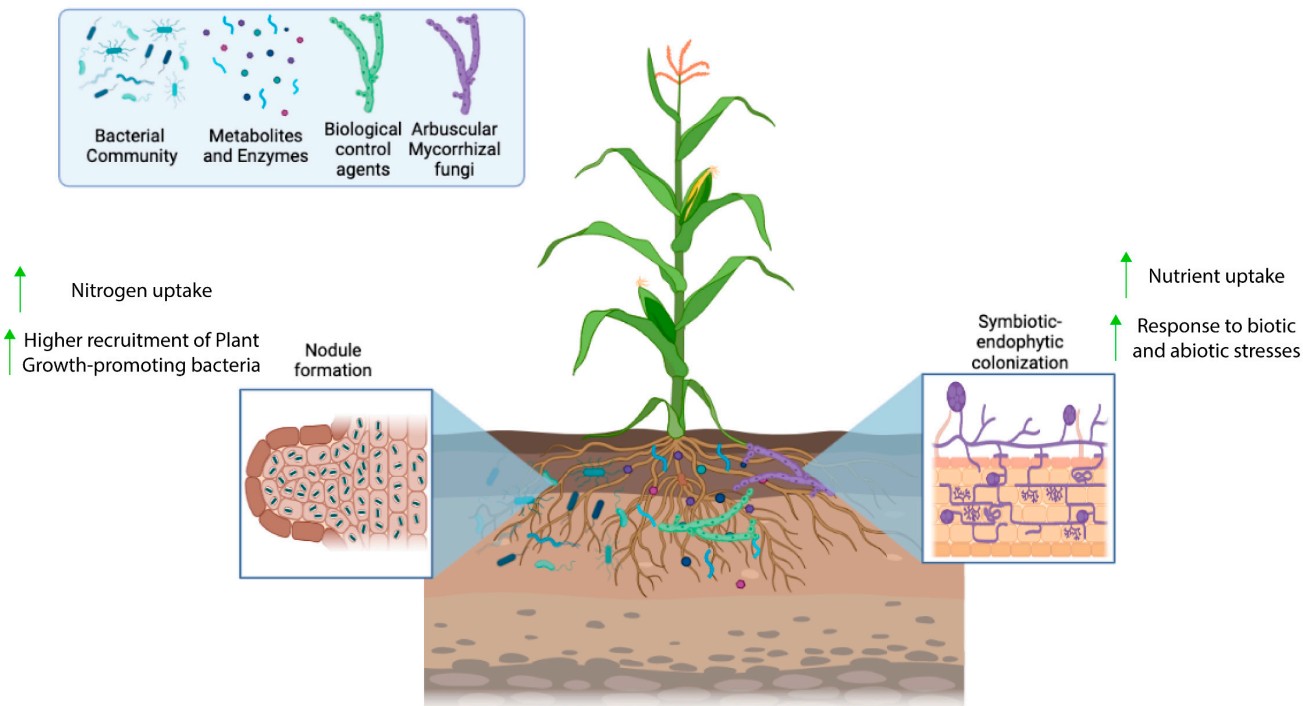

**Figure 1.** Biological processes occurring in the rhizosphere, including the growth and colonization of the plant roots by different types of microorganisms.

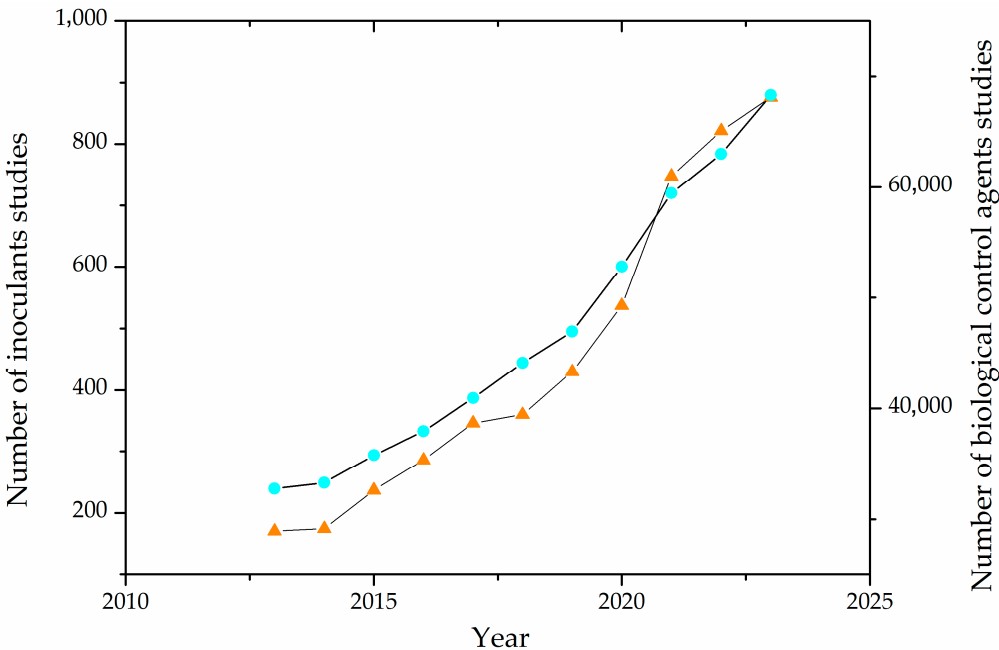

**Figure 2.** Evolution of the number of publications on agricultural inoculants and biological control agents in the last decade. The light blue circles (○) represent studies of biological control agents, and the orange triangles (Δ) represent studies of inoculants.

The main steps involved in the bioprocess to obtain inoculant formulations and biological control agents consist of the selection of suitable microbial strains; the characterization of their morphological, physiological, and biochemical properties; the evaluation of fermentation conditions (optimization of nutritional and physical parameters); and

the development of biopinputs using cheap reagents, adjuvants, and vehicles which will stabilize the bioactive agents in the formulation.

Aiming to reduce production costs, combined with the sustainable concepts of biorefineries and green and circular economies, lignocellulosic biomass has started to be used as a raw material in the development of inoculant formulations and agricultural biological control agents, being considered a viable alternative.

*Production of Inoculants and Biological Control Agents by SSF*

Inoculants had a global market share of approximately USD 1.0 to USD 2.3 billion in 2020, and their market share is expected to reach USD 3.9 billion by 2025, with a CAGR ranging between 11 and 12.8%. Biological control agents currently have a market worth approximately USD 6.6 billion, and this market worth is expected to grow at a CAGR of 15.8% to reach USD 13.7 billion by 2027. The leading global inoculant and biological control agent manufacturers are Novozymes (Frederiksberg, Denmark), BASF SE (Ludwigshafen am Rhein, Germany), Premier Tech (Boizenburg, Germany), Bioceres Crop Solutions (Rosario, Argentine), Marrone Bio Innovations Inc. (Davis, CA, USA), Bayer CropScience AG (Monheim am Rhein, Germany), and Valent Biosciences (Libertyville, IL, USA) [40,41].

With the advancements in the use of inoculants and biological control agents, the productivity of these compounds is one of the main issues that need to be evaluated for their large-scale production and global implementation. The steps that need to be taken to choose the proper microbial strain in the bioinput strikingly rely on the cultivar species, the type of disease, and the desired effect upon the system, ranging from the induction of innate plant immunity and the direct antagonism of the pathogen to the production of enzymes or improvement of soil quality. Subsequently, process conditions and substrate composition alter the dynamics of microbial metabolism; thus, the microbial strain must be cautiously selected in order to drive the production to either biomass or metabolite excretion. Both strategies contribute to the suppression of pathogens; however, the best option must be examined in situ and will depend on the prevalence of their positive effects over time or over the cultivation period.

Regarding substrate type, medium composition influences microbial growth and metabolite production. The main carbon sources assimilated by microorganisms used in inoculant and biological control formulations are sugars and polyols such as glucose, sucrose, xylose, lactose, glycerol, mannitol, and others. These nutrients can be found in different substrates. Considering the cost and availability of raw materials and substrates, agro-industrial byproducts such as lignocellulosic biomasses mainly composed of a carbohydrate portion rich in cellulose and hemicellulose stand out. As previously reported, lignocellulosic materials are promising nutritional sources that can be used in SSF for the production of bioinputs such as inoculants and control agents. In addition to supports/substrates, lignocellulosic materials can also be used as vehicles for inoculant formulations and biological control agents [42].

Zhao et al. [43] reported the production of *Rhizobium leguminosarum* biomass by SSF using fresh wheat straw and charcoal as supports. In SSF with wheat straw, it was observed that after 72 h of cultivation, there was approximately a thousand-fold increase in the number of viable cells, starting from an initial concentration of 7.3 to approximately 10.0 log cfu/g substrate. When charcoal was used, after 72 h of cultivation, a 10-fold increase in the number of viable cells was observed. The results obtained in this study showed that although charcoal is known as an excellent inert support for biofertilizer production, wheat straw presented a better performance.

*Bacillus thuringensis* is a type of bacteria present in the soil that produces certain types of proteins, known as *Cry*, with an insecticidal effect on *Lepidoptera*, *Coleoptera*, and *Diptera*; thus, it is normally used as a potential bioinsecticide. Within this context, Molina-Peñate et al. [44] reported on the use of solid waste treated with lytic enzymes (e.g., cellulases, hemicellulases, pectinases, and amylases) as raw material for the mass growth of *Bacillus*

*thuringiensis* in SSF. After 100 h of cultivation, approximately $10^8$ viable cells/g of substrate and $10^8$ spores/g of substrate were obtained. The presence of *Cry* protein crystals, essential for insecticidal action, was also observed. Accordingly, this strategy proved to be viable on the scale at which it was carried out, being an alternative process with a value-added bioinput for future biorefineries.

There are also several studies that have exploited the use of corncobs, food waste, rice husk, and coconut husks as substrates/supports for the production of cellular biomass aiming towards biological pest control [16]. Table 3 shows more examples of renewable and lignocellulosic substrates/supports used in the production of inoculants and biological control agents by SSF.

**Table 3.** Renewable substrates used in SSF for the production of sustainable inoculants and biological control agents as agricultural bioinputs.

| Substrates/Supports | Microbial Species | Yield | References |
|---|---|---|---|
| Agricultural byproducts | *Trichoderma* spp. | 689.80 mg mycelium/g substrate | [45] |
| Mixture containing rice straw | *Trichoderma guizhouense* | $4.62 \times 10^{10}$ conidia/g substrate | [46] |
| Food wastes | *Bacillus circulans* | 8–10 cfu */g substrate | [47] |
| Corncobs | *Bacillus amyloliquefaciens* | $1 \times 10^{10}$ spores/g substrate | [48] |
| Mixture containing wheat bran, rice husks, wheat straw, corn waste, corn cobs, cotton cuttings, date pits, pea hulls, potato peelings | *Bacillus sphaericus* | $1 \times 10^9$ cfu */g | [49] |
| Sugarcane bagasse | *Metarhizium anisopliae* | $5.8 \times 10^8$ conidia/g support | [50] |
| | *Fusarium caatingaense* | $1.33 \times 10^8$ conidia/g substrate | |
| Rice husk | *Beauveria bassiana* | $1 \times 10^9$ spores/g support | [51] |
| | *Trichoderma harzianum* | $1 \times 10^9$ spores/g support | |
| Wheat straw | *Beauveria bassiana* | $1 \times 10^9$ spores/g support | |
| | *Trichoderma harzianum* | $3.0 \times 10^9$ spores/g support | |
| Orange peel | *Beauveria bassiana* | $7.0 \times 10^7$ spores/g support | |
| | *Trichoderma harzianum* | $8.0 \times 10^9$ spores/g support | |
| agricultural digestate from anaerobic biogas production mixed with apple, banana and grape wastes | *Trichoderma reesei* | 700 mg/g substrate | [52] |
| | *Trichoderma atroviride* | 700 mg/g substrate | |

* cfu: colony-forming unit.

Some authors also consider inoculants and biological control agents as biofertilizers, as in addition to microorganisms, these formulations can also contain substances such as organic acids, amino acids, vitamins, enzymes, micronutrients, and phytohormones, which promote plant growth [53]. Thus, in Table 4 these mentioned substances are classified according to their function upon the stimulation of growth.

**Table 4.** Classification of bioinputs according to their major functions as exhibited in inoculants and biological control agents.

| Substance | Function | Reference |
|---|---|---|
| Organic acids | Substances that promote reductions in soil pH by solubilizing phosphates that will be assimilated by plants later. | [54] |
| Phytohormones | Compounds such as gibberellin, indoleacetic acid (IAA), cytokinins, salicylic acid, auxin, and ethylene are synthesized by some microorganisms and act in the development of the root system, as well as in the processes of pollination, germination, growth, flowering, fruiting, and plant defense. | [55] |
| Amino acids | Substances that act in the production of proteins, as well as signal molecules, regulating root and shoot architecture and regulating flowering time and stress defense. | [56] |
| Enzymes | Some hydrolases produced act in the biological control of pests and in the decomposition of organic matter present in the soil aiming at plant nutrition. | [57] |
| Vitamins | These compounds are important regulators of cellular metabolism in plants; they act as cofactors in many enzymatic reactions, as well as antioxidants. | [58] |
| Micronutrients | Elements that can act as enzymatic cofactors, influencing plant immunity, and against abiotic stress. | [59] |

In addition to the studies cited in Table 3, there are also works that describe the production of biofertilizers using lignocellulosic residues under anaerobic digestion (AD) to generate biogas [60]. In AD, organic matter is degraded and stabilized by microbial action in an anoxic environment, generating biogas as the main product [61]. At the end of AD, organic acids (acetic acid, propionic acid, butyric acid, caproic acid, and valeric acid), solvents (acetone), and digestate (solid material resulting from the process) are also observed as byproducts [62]. The digestate can be used as a biofertilizer for agricultural crops, replacing mineral fertilizers. The application of digestate as a biofertilizer depends on its physicochemical composition. Its composition is highly variable depending upon the feedstock, pretreatment process, inoculums used, and AD operating conditions like pH and temperature. Commonly, it is characterized by an alkaline pH; $N–NH_4^+$ (46.2–79%); macronutrients such as potassium (28–95 g/kgDM), phosphorus (8–42 g/kgDM), sulfur (2.9–14.7 g/kgDM); and trace elements such as cobalt, iron, and selenium [63].

Regarding the production of biofertilizers, Vassilev et al. [64] demonstrated the feasibility of using solid beet residues for the growth of microbial biomass of *Aspergillus niger*, which, by producing organic acids during the fermentation process, solubilized the natural phosphate, making it available for the plant.

Ajala et al. [65] showed that *A. niger*, when cultivated on cassava peel by SSF, produces a concentration of citric acid three-fold higher in comparison to SmF. Devi and Sumathy [66] evaluated the production of biofertilizers in SSF using different fruit residues (papaya, watermelon, pineapple, custard apple, and guava). Microorganisms isolated from soil samples were evaluated, and the growth of *Pennisetum glaucum* was observed upon their application. After microbial growth, the plants were inoculated with fermented custard apple, guava, and watermelon. Greater seed germination and more pronounced elongation regarding the roots and shoots were observed in the plants treated with fermented solutions compared to the control plants.

Solid biofertilizers obtained by SSF from lignocellulosic biomass when applied to the soil are also important for the formation of humic substances. These substances have long been recognized as the most widely distributed organic component on the planet. Humic substances are formed mainly from the biological degradation of plant and animal residues. Their structures have an abundance of carbonyl and phenolic groups that contribute to

their complexation and have amphipathic characteristics that can bind to mineral soil surfaces [67–70]. The main types of humic substances found in soil are fulvic acids, humins, and humic acids, which act at different pH. Humic substances contribute to heat retention in the soil, which stimulates seed germination and plant root development; they act against erosion, preventing runoff due to the fact that they contain aggregates from combining with clays, enabling a better water retention capacity [71]. When the solid biofertilizer obtained by SSF, still containing a significant lignin content, comes into contact with soil microorganisms such as rhizospheric bacteria, it will be degraded by the action of phenoloxidases and enzymes that produce hydrogen peroxide. The resulting phenolic compounds will be precursors of humic substances. Some studies have demonstrated the efficient production of humic acids by SSF using lignocellulosic agro-industrial byproducts such as raw empty fruit bunch and rice straw fibers [72,73].

In addition to lignocellulosic substrates, other materials can be used in the production of inoculants, biological control agents, and biofertilizers. Examples include cheese whey, beetroot and sugarcane molasses, glycerol, solid residues from olive oil mills, and vinasse, which can be used in SSF as a substrate/support or moistening solution [21,64,74,75].

In addition to inoculants and biological control agents, other bioinputs such as biosurfactants and phytohormones can also be produced by SSF, as reported in the next sections.

## 4. Biosurfactants: Promising and Emerging Agricultural Bioinputs

Biosurfactants (or biological surfactants) are microbial metabolites with an amphipathic structure that have outstanding physicochemical and biological properties such as the ability to reduce surface tension, emulsifying capacity, and antimicrobial potential. According to the biochemical groups present in their structures, biosurfactants can be classified as glycolipids, lipopeptides, lipoproteins, phospholipids, and similar molecules [76]. Glycolipids and lipopeptides are the most common microbial biosurfactants.

Glycolipids are formed by the union of sugars (monosaccharides and polyols) and lipids, and they have molar masses between 476 and 848 Da and critical micellar concentrations (CMCs) between 20 and 366 mg/L [77,78]. Among the most common biosurfactant glycolipids are rhamnolipids, produced by bacteria, and sophorolipids, produced by yeast.

Lipopeptides are biosurfactants formed by the union between peptides and lipids and have molar masses greater than 1000 Da; due to this, they are called high-molar-mass biosurfactants, and they have CMCs around 10 μmol/L or 23 mg/L. Surfactin is the lipopeptide produced by some bacteria best described in [77].

In agriculture, microbial biosurfactants are commonly used as emulsifiers in agrochemical formulations. However, in recent years, with the search for "clean" methodologies for more sustainable agricultural production, it has been observed that biosurfactants can also be used as antimicrobials, germinating agents, and growth stimulants.

Due to their antimicrobial properties, in recent years biosurfactants have been the target of agronomic studies aiming to use them as biopesticides to control pathogens in crops. The antimicrobial activity of biosurfactants is due to their membranotropic effects. Upon contact with pathogen cells, biosurfactants cause the permeabilization of the target cell through interacting with lipids and membrane proteins, causing cell disruption and death. The interaction mechanisms of the interactions between biosurfactants and membrane molecules involve three processes: (I) a change in membrane hydrophobicity through the intermolecular interactions between different functional groups present in biosurfactants, lipids, and proteins [79]; (II) the fluidization of the cell membrane, which alters the balance between anchored saturated and unsaturated fatty acids [80,81]; (III) increasing the exchange of charge between the anchored molecules and thus opening pores and channels that, in turn, favor greater permeabilization of the cell membrane [82, 83]. In turn, these aforementioned mechanisms cause the efflux of bioactive intracellular compounds such as enzymes, lipopolysaccharides (LPS), polypeptides (PP), and functional effectors, namely Microbial Associated Molecular Patterns (MAMPs), which could stimulate or "sensitize" plants, leading to a faster response of systemic resistance to pathogens [84].

Still, regarding their agricultural applications, there are also reports that biosurfactants produced by some endophytic microorganisms act as protective molecules, acting significantly on growth and against some biotic and abiotic factors [85,86].

*Production of Biosurfactants in SSF*

Despite being molecules of fundamental importance for the industrial and agricultural sectors, the production costs of biosurfactants are still high. This fact is due to the raw materials used and the purification processes adopted [86]. In recent years, the use of agro-industrial byproducts for the production of biosurfactants has increased. Various lignocellulosic materials can be used in the production of biosurfactants via SSF, as can be seen in Table 5.

**Table 5.** Different approaches for the production of different types of biosurfactants via SSF using renewable feedstocks.

| Substrate | Microorganism | Biosurfactant Type | Yield | Reference |
|---|---|---|---|---|
| Rice Straw and non-fat Rice bran | *Aspergillus fumigatus* | NI * | 8.47 EU */g | [87] |
| Tuna fish residue flour and potato peel flour | *Bacillus subtilis* SPB1 | Lipopeptide | 27.1 mg/g dry substrate in 48 h | [88] |
| Sunflower seed oil | *Pleurotus ostreatus* | Carbohydrate-peptide-lipid complex | 4.69 g/L in 20 days | [89] |
| Rice straw | *Bacillus amyloliquefaciens* XZ-173 | Surfactin | 15.03 mg/g dry substrate in 48 h | [90] |
| Multiple agricultural and food wastes | *Bacillus subtilis* SPB1 | Surfactin | 20.8 mg/g dry substrate in 48 h | [91] |
| Stearic acid and molasses | *Starmerella bombicola* ATCC 22214 | Sophorolipid | 211 mg/g substrate in 16 days | [92] |
| Peanut oil cake | *Bacillus cereus* SNAU01 | Lipopeptide | NQ * | [93] |
| Pretreated molasses | *Brevibacterium aureum* MSA13 | Lipopeptide | 18 g/L in 7 days | [94] |
| Olive leaf residue flour and olive cake flour | *Bacillus subtilis* SPB1 | Lipopeptide | 30.67 mg/g solid material in 48 h | [95] |
| Mahua oil cake | *Serratia rubidaea* SNAU02 | Rhamnolipids | NQ * | [96] |
| Okara and sugarcane bagasse | *Bacillus pumilus* UFPEDA 448 | Surfactin | 1156 mg/L impregnating solution | [97] |

* NI: not identified; EU: emulsifying units; NQ: not quantified.

As previously reported, obtaining biosurfactants by SSF is still carried out on small scales, with only 2% of studies reporting a yield of between 5 and 10 kg of bioproduct. Furthermore, 62% of the studies on obtaining biosurfactants by SSF reported the production of glycolipids (sophorolipids, rhamnolipids, and others) and lipopeptides (surfactin, iturin, and others), and 40% of the studies reported the use of packed beds or porcelain fully aerated and agitated bioreactors [98].

When analyzing the advantages of obtaining biosurfactants by SSF when compared to SmF, a greater production of biosurfactant by SSF is observed. However, disadvantages such as the complexity in establishing mathematical models, as well as the difficulties in purification and scaling up, show that there are still many advances to be achieved in this promising bioprocess for industrial application, considering sustainable concepts for a green economy.

Glycolipids are a type of biosurfactant with antifungal, larvicidal, and mosquitocidal properties, thus acting as agricultural defenders against pests and fungal infections [99,100]. Glycolipid-based biopesticides are applied to agricultural crops to control phytopathogenic bacteria, fungi, and pests [101].

The antibiotic properties of biosurfactants have been explored for some time and are well known. Antifungal properties stand out, since there is a more limited number of active ingredients for the treatment of these phytopathogens when compared to antibiotics.

It is known that fungi such as mold/white rot (e.g., *Sclerotinia sclerotiorum*, *Penicillium digitatum*, *Penicillium italicum*) and black rot (e.g., *Alternaria citri*), among others, cause major losses for leguminous crops such as soybeans, beans, and citrus fruits. The routine use of a variety of synthetic fungicides, such as triazoles, triazolinthione, carbamates, dithio-carbamates, organochlorines, and organophosphates, has caused several environmental problems, such as the pollution of soil, groundwater, and water sources. Therefore, the adoption of biosurfactants with fungicidal effects proves to be a sustainable alternative for agricultural production.

Hultberg et al. [102] demonstrated the antifungal effect of biosurfactants produced by *Pseudomonas fluorescerans* against *Pythium ultimum* (causative agent of the damping off and root rot of plants), *Fusarium oxysporum* (causes wilting in crop plants), and *Phytophthora cryptogea* (causes rotting of fruits and flowers). *Pseudomonas* sp. have been reported as biocontrol agents against *Verticillium microsclerotia*, a causative agent of *Verticillium* wilt mainly in potatoes. The biosurfactant produced by this *Pseudomonas* sp. is considered to play a major role in the inhibition of the in vitro viability of *Verticillium* sp. [103].

Nalini and Parthasarathi [96] demonstrated the action of rhamnolipids produced by *Serratia rubidaea* SNAU02 as a biocontrol agent. In this study, the authors evaluated the use of mahua cake as a substrate for biosurfactant production via SSF. The rhamno-lipid produced showed antifungal activity against *Fusarium oxysporum* and *Colletotrichum gloeosporioides*. The study suggested that this biosurfactant forms ion channels in the plasma membrane of the microorganism, creating pores in the membrane layer that affect the cell surface of plant pathogenic fungi.

The action of a rhamnolipid produced by *Pseudomonas aeruginosa* Tr20 in combination with a compatible strain of *Trichoderma lixii* TvR1 was also evaluated for the control of fusariosis and early blight in plants. Through these in vivo activities against the pathogens *Fusarium oxysporum* and *Alternaria solani*, a significant reduction in the incidence of fusario-sis and early blight by 27.21% and 29.43%, respectively, was observed [104].

Lipopeptides are also important bacterial biosurfactants with agricultural applications. According to Hoff et al. [105], surfactin secreted by *Bacillus velenzis* in contact with plant roots acted as a bioactive secondary metabolite (BSM), acting as a signal and/or antimi-crobial and being responsible for optimizing biofilm formation, motility, and early root colonization and increasing plant immunity.

With the COVID-19 pandemic, studies on the antiviral potential of biosurfactants increased [100,106–109]. In the agricultural sector, the antiviral effect of these molecules is of fundamental interest, since they can be used against phytopathogens such as Potyvirus (family *Potyviridae*), Tospovirus (family *Bunyaviridae*), Tobamovirus (family *Virgaviridae*), Closteovirus (family *Closteoviridae*), Begomovirus (*Geminiviridae*), and Polerovirus (family *Luteoviridae*), which affect crops of economic importance such as vegetables, soybeans, beans, sugar cane, potatoes, and citrus fruits [110–115]. In viruses, biosurfactants can act by disintegrating particles through interaction with lipid layers and denaturing proteins in structures such as the capsid and envelope [107].

In addition to their effects against pathogens, biosurfactants are also responsible for the disruption of the waxy cuticles of some insects [99]. The removal of these lipid structures can cause dehydration in the insect and also make it more susceptible to infection by biological control agents and the action of hydrolase enzymes, facilitating the death of the insect pest.

## 5. Phytohormones as Agricultural Bioinputs

### 5.1. Gibberellic Acids

Gibberellins, or gibberellic acids (Gas), are a family of diterpenoid acid tetraclics biosynthesized by plants, fungi, bacteria, and some algae genera that present hormonal effects in plant growth at low concentrations. These biomolecules exhibit structures with carboxylic acid, the presence or absence of unsaturated ring bonds, and different hydroxy-lation degrees; they have a molar mass of around 346.38 g/mol, are soluble in water (4.6

g/L in the room temperature), and have a melting point between 223 and 225 °C. Currently, 136 types of GA have been identified, and $GA_1$, $GA_3$, $GA_4$, $GA_5$, $GA_6$, and $GA_7$ are considered the most active. Their typical structures are represented in Figure 3 [116,117].

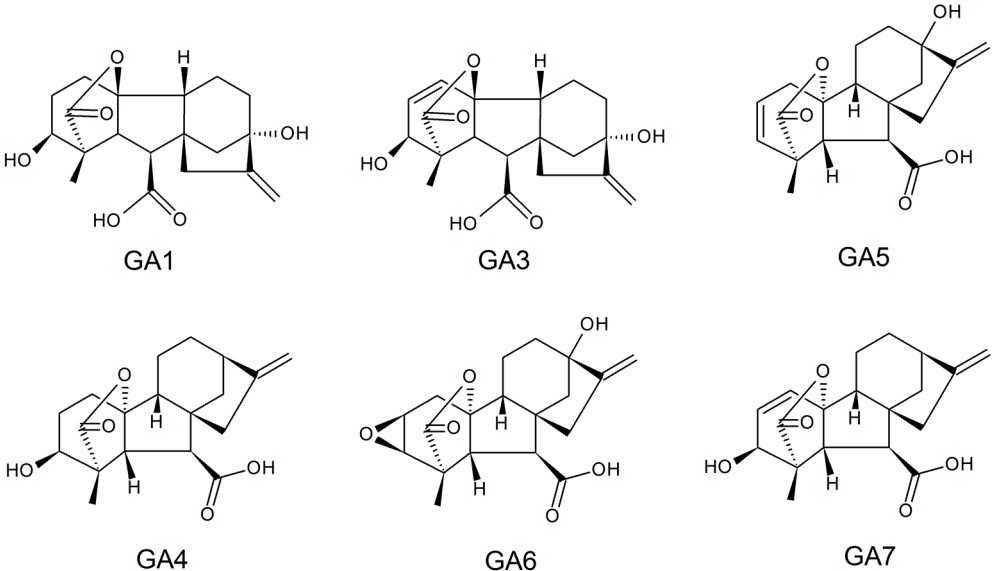

**Figure 3.** Typical active structures reported for gibberellin.

GAs are important in plant physiology because they can promote stretching [118], germination and seed dormancy [119], flowering [120], gender development [121], leaf inhibition and senescence [122], and fruit aging [123]. Non-active forms of GA do not accumulate in plant tissues or cells but are rather transported in a root-to-shoot manner, or vice versa [124]. On the other hand, bioactive forms of GA such as $GA_4$ accumulate in the sepals and petals during organ development [125]. Moreover, one study showed that GA levels are higher in the root elongation zone compared with the meristematic zone, pointing towards the hypothesis that GAs are synthesized locally or even transported from the surrounding tissues to compensate for hormone dilution during cellular growth [126,127].

Currently, GAs are bioinputs that have expanded the market in recent years, having been mainly used in horticulture [128–131]. Recent studies also show that gibberellin has potential applications as a biostimulant related to the main agricultural products of developing countries like corn [132,133], soybeans [134,135], sugarcane [136,137], and cotton [138]. Some studies have also shown that GA formulations (granular, pure, mixtures, and others) are used to protect seeds, flowers, and fruits against larvae and insects and that they act as herbicides [139,140].

According to the strategic importance of these bioinputs for achieving increasingly sustainable agriculture, the market trends show that production and profit are in ascendancy. The global market for GAs was valued at USD 500 million in 2017, and it is expected to reach revenues of 1167.8 million dollars in 2025, with an annual increase of 8.8% in CAGR [141]. The countries with the largest GA-producing companies are China, USA, and New Zealand.

Although the use of GAs is still restricted to small- and medium-sized plantations, the adoption of these bioinputs for farmers in developing countries, considered as "world granaries", could further boost their use. Therefore, it is necessary to achieve high-yield production processes that use sustainable technologies and renewable raw materials such as lignocellulosic biomasses.

Currently, the industrial production of GAs has been predominantly carried out by SmF. However, diluting the sample in the fermentation broth requires a later concentration step, which makes the process more expensive [142]. It is also possible to produce GAs by chemical synthesis or by extraction from plants, but these methods are not economically

viable. Thus, in recent years, the microbial production of GAs in SSF has presented itself as economically viable, whereby the main feedstock are agro-industrial byproducts [143]. In addition, in GA production via SSF, there is no concern with catabolic repression owing to the fact that carbon sources are present in solid state, majorly as carbohydrates; therefore, the microorganisms can possibly have better control over the available nutrients that can be absorbed by the excretion of cellulolytic enzymes.

GAs are produced via SSF generally using filamentous fungi as fermenting agents, the most common of which being *Fusarium moniliforme* and *Gibberella fujikuroi* [144]. In addition to filamentous fungi, bacteria can also produce GAs, especially those that promote plant growth and fulfill the role of nitrogen fixers, such as *Azotobacter chroococcum*, *Bacillus licheniformis*, *P. fluorescens* and *Azospirillum lipoferum* [144–146]. The use of bacteria that promote plant growth in the production of GAs comes with the advantage of obtaining a product for multifunctional agricultural application, acting as a biofertilizer and stimulant/regulator of plant growth [144].

One important strategy for discovering new GA-producing microorganisms is the isolation of endophytes. These microorganisms symbiotically inhabit plant tissues, and their metabolites usually foster plant growth. Studies have pointed to *Fusarium oxysporum* strain and *Porostereum spadiceum* [134,147].

Regarding the selection of the feedstock, primarily, the water activity ($a_w$) of the raw material must be considered for the survival of the selected microorganism. For instance, in *Gibberella fujikuroi* grown via wheat bran and soluble starch, it was observed that the minimum $a_w$ bore by the microorganism was 0.9 $a_w$, whereas optimal growth and GA production was achieved between 0.985 $a_w$ and 0.995 $a_w$ [148]. Nonetheless, in the literature, there is still a lack of studies on GA production using yeast under SSF and controlling $a_w$. In general, it is known that filamentous fungi thrive on substrates with a relative humidity between 50 and 70% (commonly used for SSF). Bacteria and yeasts require higher levels of moisture to grow. However, the elevated moisture content in the substrate hinders oxygen diffusion through the solid bed and facilitates contamination by other microorganisms, which, in turn, interferes with the fermentation performance. In this context, the use of filamentous fungi for the production of GAs via SSF is preferred.

Analogously, the nutritional elements and nature of the culture medium are also fundamentally important parameters for GA production. The main carbon sources used in GA production are known to be carbohydrates and polyols such as glucose, sucrose, galactose, mannitol, maltose, starch, and glycerol [149–153]. These substrates are found in large quantities in lignocellulosic and starchy-derived materials such as corn stalks, citric peel, wheat bran, sugarcane bagasse, citric pulp, soy bran, sugarcane bagasse, soy husk, cassava bagasse, and coffee husk [117].

Production of Gibberellic Acids via SSF

Table 6 shows some raw materials used for GA production via SSF.

The use of agro-industry byproducts to obtain high-value-added products has been highlighted, and GAs could be alternatives to products in biological refineries.

N sources, mineral salts, and trace elements are also important in obtaining GAs. The use of agro-industrial byproducts with complex compositions dismisses supplementation with sources of N, mineral salts, vitamins, etc. If the addition of N and mineral salts is necessary, the use of liquid residual extracts such as corn steep liquor, plant meals, rice bran, or soy extract may be a viable and cheaper option than the addition of inorganic nitrogen salts solutions. It is interesting to use them as moisturizers, thus controlling the humidity of the process and supplementing it with nutrient sources. Another advantage of using organic N sources in fermentation is that during metabolism, the degradation of nitrogen compounds (hydrolysis reactions) will not drastically affect the pH. Due to these characteristics, complex organic nitrogen sources are favorable for the production of GAs.

**Table 6.** Different residual substrates used in the production of gibberellins via SSF.

| Microorganism | Substrate | Yield | Reference |
|---|---|---|---|
| *Fusarium moniliforme* | Wheat bran and soluble starch | 1.16 g/kg of dry substrate | [154] |
| *Fusarium moniliforme* LPB 03 | Citric pulp and soy husk | 5.9 g/kg of dry substrate | [155] |
| *Gibberella fujikuroi* | Rice bran and barley malt residue | 10.1 g/kg dry substrate | [156] |
| *Paecilomyces* sp. ZB | Cow dung | 1.31 g/kg dry substrate | [157] |
| *Fusarium moniliforme* CICC | Enzymatic hydrolysate corn stalks | 9.48 g/kg dry substrate | [158] |
| *Gibberella fujikuroi* | Mixture of rice bran and malt residue | 1.3 g/kg dry substrate | [159] |
| *Fusarium Oxysporum* | Mixture of sesame bark and wheat straw residue | 7.14 g/kg dry substrate | [147] |

However, even with the wide availability of N sources (e.g., organic and inorganic forms), it is not usual to vary nitrogen concentration while SSF occurs due to the difficulty of the operation, which makes it difficult to homogenize the system. Nonetheless, Panchal and Desai [154] evaluated the effect of supplementary N sources (e.g., $NH_4Cl$, $NH_4NO_3$, $(NH_4)_2SO_4$, and urea) in commercial wheat bran (CWB) fermentation for the production of $GA_3$ under SSF with 70 mg of N (differing only in the nature) per 100 g CWB. The utilization of urea resulted in an enhancement in the productivity of the GA. Indeed, $GA_3$ production was higher (234, 489, and 532 µg/g) compared to the assay without N supplementation (95, 125, and 163 µg/g) at 120, 144, and 168 h, respectively.

Parameters such as pH and temperature are also important factors in SSF for GA production. Controlling the pH in SSF for GA production is difficult, but this problem can be solved by the addition of buffers or solutions with pH between 3.5 and 5.8. Satpute et al. [143] tested SSF using pigeon pea at a pH range between 4.3 and 5.3, and it was shown that the optimal production of $GA_3$ occurred at pH 4.76, probably because of the growth of the fungus. On the contrary, Rodrigues et al. [142] established an initial pH between 5.5 an 5.8, which corresponds to the natural pH of citric pulp used as a substrate/support for GA production. This strategy represented an interesting approach to lowering the cost of production of this bioinput.

On the other hand, Bai et al. [158] used enzymatic hydrolysate from steam-exploded corn stalks to produce GAs. The pH of the enzymatic hydrolysis employed in this study was 4.8; however, the pH of the biosynthesis process was 6. In this case, the temperature range used in SSF for GA production is between 25 and 34 °C, because the fungi used as fermenting agents are mesophiles. Satpute et al. [143] described the fermentation of pigeon pea pods at various temperatures (e.g., 20 °C, 29 °C, 37 °C, and 45 °C) and demonstrated that the best temperature for the production of gibberellin was 29 °C, which is very close to the ideal range of spore germination and growth and product formation in SSF using *Fusarium proliferatum*.

Aeration in SSF for GA production is only controlled via bioreactors. This parameter is important because GA biosynthesis is a process with several oxidative steps. Thus, the aeration control is critical for the optimization of the bioprocess. The effect of aeration was observed by Oliveira et al. [117], who compared the production of GAs in SSF using Erlenmeyer flasks and a column bioreactor with forced aeration (30 mL/min). In addition to having a higher productivity (in 120 h of fermentation), production with forced aeration led to a GA gain of 13.72% (*w/w*) in relation to the results obtained using the Erlenmeyer flasks. The granulometry of the substrates used in SSF can also contribute to aeration. It is known that substrates with a small particle size can cause the "packaging effect" in the bioreactor, making oxygenation difficult. However, the use of substrates with larger particle sizes does not favor the enzymatic process that occurs in SSF due to their large

contact surfaces. Thus, a valid strategy is to mix substrates with different particle sizes; this strategy was used by Rodrigues et al. [142], who tested sizes between 0.8 and 2 mm in mixtures of different substrates (citric pulp, soy bran, sugarcane bagasse, soy husk, cassava bagasse, and coffee husk).

*5.2. Auxins*

Auxins are compounds recognized as a family of aromatic molecules with a carboxylic acid group. They can be synthesized by plants and microorganisms and are responsible for controlling the plant growth response and development by means of cell differentiation, division, and expansion at low concentrations [160,161]. Commonly, these biomolecules are found in natural forms, as exemplified by indole-3-acetic acid (IAA), 4-Cl-IAA, phenylacetic acid (PAA), indole-3-butyric acid (IBA), and indole-3-propionic acid (IPA), and as exhibited in Figure 4 [161,162].

**Figure 4.** Main active auxin forms found in plants.

There is also the occurrence of other types of auxins such as auxin a (auxenotriolic acid), auxin b (auxenolonic acid), benzofuran-3-acetic acid (BFA), and phenyl-butyric acid (PBA) in plants or other organisms, which is still being debated by experts or not fully understood. Also, synthetic auxins exist as active 2,4-dichlorophenoxyacetic acid (2,4-D) and naphthalene acetic acid (NAA), as well as the inactive precursor 2,4-dichlorophenoxybutyric acid (2,4-DB) [162]. In addition, auxins are hormones that partly regulate the processes of vascular tissue development, including tissue regeneration and vascular tissue connection, both of which are important for graft union [163].

Over the years, many studies have shown that auxin-regulated cell expansion plays an important role in plant development, specifically in primary root growth [164,165], lateral root development [166,167], organ development [168,169], tropism [170,171], apical dominance [172,173], etc. Because of their multifunctionality, auxins can be used in strategic plantations as plant growth promoters, and this approach has shown promising results in sugarcane [174], potato [175], and tomato plantations [176]. Auxins also help to obtain higher yields of good quality carrot seeds [177], improve the root morphology and growth of grafted cucumber seedlings [178], and can be used to alleviate abiotic stresses [179–181].

The most well-known compound used to treat weeds using auxin-based products in plantations is 2,4-dichlorophenoxyacetic acid (2,4-D), which was discovered in the 1940s and has been used extensively for more than 70 years worldwide. Arylex™ (developed mainly for cereal plantations) and Rinskor™ (used in rice fields) are new types of auxin herbicides produced on the market which have proven to be an attractive alternative to auxin application [182].

Due to studies carried out over the years showing the efficiency of using auxins in crops and improving their production, using this bioinput has been an important strategy

for producers and their productions [177,178]. The production of auxins by chemical companies and laboratories started with the development and use of biostimulants with other phytohormones. Despite their great advantages, currently, synthetic auxins are largely employed rather than biosynthesized ones. Due to the environmental appeal and the commercial requirement for biodegradable products, auxin-producing microorganisms like fungi, bacteria, and algae are increasingly being researched and studied. The most well-known auxin of natural origin is indole-3-acetic acid (IAA), which has been widely studied in the literature. IAA is a molecule mainly produced by plants and microorganisms associated and not associated with plants. In the context of plant–microbe interactions, the presence of IAA plays a signaling role and helps to direct the growth and development of the plant [183].

In general, some Tryptophan (Trp)-independent routes have been described in the biosynthesis of IAA. For instance, a Trp-independent pathway that produced 90% of the IAA in the absence of tryptophan was observed by Prinsen et al. [184] in the nitrogen-fixing rhizobacteria *Azospirillum brasilense* Sp6 and its Tn5 mutant SpM7918. In addition, Li et al. [185] suggested the presence of an unknown tryptophan-independent pathway for IAA production by *Arthrobacter pascens* ZZ21 through endogenous precursors. Moreover, Roesch et al. [186] screened several microorganisms in order to discover promising *Azospirillum* spp. to promote plant growth through nitrogen fixation and auxin production in the southern fields of Brazil. Markedly, 30 isolates from rhizospheres, roots, and stems of maize were identified and evaluated as great candidates to synthesize auxin in the absence of tryptophan supplementation. However, nitrogen played a crucial role in regulating auxin pathways once the N-free media showed no trace of the product.

Suggestively, by comparing other studies regarding the isolation of potential auxin-producing bacteria such as *Bacillus*, *Pseudomonas*, *Escherichia*, *Micrococcus*, and *Staphylococcus*, it was presumed that bacteria that have strengthened N-fixing ability and are responsible for nitrogen cycling have more versatility upon auxin biosynthesis and may present more alternative routes to produce auxins [187]. Further investigations are necessary to elucidate the mechanism and cofactors required to obtain consistent and sufficient results.

On the contrary, Trp-dependent routes have been more frequently reported in the literature. For example, the indole-3-pyruvic acid (IPyA) pathways in some bacteria and fungi have been studied. This route involves the presence of decarboxylases to convert IPyA to indole-3-acetaldehyde (IAAld), which is oxidized to IAA by a dehydrogenase [188]. On the other hand, in the first step of the indole-3-acetamide (IAM) pathway, tryptophan is converted to IAM by tryptophan-2-monooxygenase. Subsequently, IAA is synthesized by IAM hydrolase from IAM. Interestingly, the genes found in plants share similarities with the enzymes that convert IAM to IAA in bacteria, which suggests the common evolution of this auxin biosynthesis pathway in bacteria and plants [189,190]. Moreover, the tryptophan side-chain oxidase (TSO) route indicates that the tryptophan is converted to IAAld, followed by the oxidation of IAAld to IAA. The decarboxylation of tryptophan to tryptamine (TAM) occurs in the TAM pathway. Then, an amine oxidase catalyzes the conversion of TAM to IAAld, which is oxidized to IAA by oxidation. In the case of the indole-3-acetonitrile (IAN) pathway, IAA can be obtained from the direct conversion of the intermediates by nitrilase or the conversion of IAM to IAA by nitrile hydratase [185,191]. For detailed information of the possible pathways currently available in the literature, we recommend Di et al.'s review [192].

Within this context, it is important to investigate distinct routes and means to synthesize IAA that function independently of tryptophan, broadening the spectrum of alternatives to afford higher productivities regarding IAA and making its industrial implementation economically viable. Regarding the already identified pathways of auxin-producing microorganisms, it is still necessary to conduct research aimed at unraveling the parameters and alternative routes for the synthesis of auxins as well as understanding the metabolism of auxin regulation, seeking high productivity, preferably through using byproducts from renewable sources as raw materials.

Production of Auxins in SSF

Despite its immense advantages in the use of SSF, the production of auxin as IAA using this fermentative method has been underexplored. However, Prado et al. [193] screened microorganism strains like *Aspergillus* spp., *Bacillus* spp., and *Trichoderma* spp. to investigate their capability to produce auxin (indole-3-acetic acid) and phytases through SSF in different substrates followed by the optimization of the process. The study revealed that all strains can produce auxin (comprising higher productivity with tryptophan supplementation). The highest indole derivative levels were observed in wheat bran as substrate by *B. subtilis* D strain (158 µg/mL), but the same strain did not produce auxins in cassava bagasse. These findings led the authors to investigate the physico-chemical characteristics of each substrate to set a reliable framework of the reasons for that response.

Presumably, the high starch and lower protein content could hinder auxin synthesis due to the N-limiting condition. Additionally, due to the fact that cassava bagasse was the substrate with the highest lignin composition and as indicated by the results, lignin has a strong negative correlation with the production of IAA, observed in the *Bacillus* and *Trichoderma* strains. Indeed, lignin stands as a physical barrier hampering the microbial and enzymatic access to the carbohydrate fraction of the biomass. In this regard, the selective degradation of lignin to carbon dioxide can be carried out using white rot fungi. For instance, Giri and Sharma [194] suggested pretreating wheat straw using *Phanerochaete chrysosporium*, known for its ability to degrade lignin, as an alternative to accessing cellulose and converting it into easily fermentable sugars. Nevertheless, during secondary fermentation, these precursors may contribute to the synthesis of IAA via this fungus, along with the supplementation of tryptophan.

Another study conducted by Prado et al. [195] evaluated the production of indole-3-acetic acid and its derivatives in 16 microbial strains. For the first time in the literature, this study reported high indole-3-acetic acid production using strain *Aspergillus flavipes* (ATCC® 16814™) through a tryptophan-dependent pathway in SSF, utilizing an optimized medium culture composed of soybean bran and tryptophan that achieved a yield of 71 ug mL$^{-1}$. The highest indolic compound yield was 655 ug mL$^{-1}$ using soybean bran as a substrate and the following conditions: particle size >1 mm and 15 mL of water supplemented with 1.5% of tryptophan. The fermentation broth was mixed in solid form (SF) and liquid form (LF) and used in the cultivation of hybrid *Eucalyptus grandis* and *E. urophylla* (clone IPB2). In response to the treatment, the plants showed an improvement in adventitious rooting rate, root length, both root fresh and dry mass, and root length and dry mass, respectively, 40 days after planting. Also, the plant biostimulant was confirmed to be non-toxic and eco-friendly by comparing the viability of (NIH/3t3) in the plants' fibroblasts cultivated with indole-3-acetic and the control group (without exogenous phytohormone application). Another advantageous aspect of SSF is that the product can be applied as a compost directly in the soil, allowing it to be a promising technique for agriculture.

One of the major drawbacks of producing IAA in large quantities in fermentations is the dependence on supplementing the culture medium with tryptophan. The addition of tryptophan increased production by, on average, 10-fold in SSF involving tryptophan in almost all strains tested in one study carried out by Prado et al. [193]. Further, this trend was also observed in another study by Prado et al. [195] after optimizing the culture medium using *A. flavipes* as an IAA producer microorganism. The authors detected a 1716-fold increase in IAA production under SSF compared to the assays absent of tryptophan.

Although the product generated has huge potential for use in the formulation of bioinputs, the supplementation of the amino acid tryptophan to form IAA makes the production of this hormone more expensive, especially on an industrial scale. One of the possible solutions to this problem is the use of raw materials that contain tryptophan in a nutritional form, such as wheat or green pea, which contain 0.125 (g/g dry mass) and 0.180 (g/g dry mass), respectively [196].

Another alternative is the intensification of IAA synthesis in microorganisms through metabolic engineering. For instance, Romasi and Lee [197], cloned *E. coli* aminotransferase,

indole-3-pyruvic acid decarboxylase, and indole-3-acetic acid dehydrogenase, which are three enzymes involved in the indole-3-pyruvic acid pathway. The results showed that the recombinant *E. coli* DH5$\alpha$ produced about 1.1 g/L of IAA after 48 h of cultivation in LB medium supplemented with 2 g/L of tryptophan. However, after the deletion of a gene that mediates indole formation from tryptophan, the yield of IAA obtained from the new strain *E. coli* IAA68 was 1.8 g/L of IAA, which corresponded to a 1.6-fold increase when compared to wild-type DH5$\alpha$.

In contrast, Malhotra and Srivastava [198] expressed a heterologous indole acetamide IAM pathway consisting of the iaaM and iaaH genes in *Azospirillum brasilense* SM. Subsequently, the IAA levels increased by three-fold, and the sorghum roots as well as the dry weight of the plants improved when compared to the effect of the wild-type strain. Similarly, Kochar et al. [199] overexpressed the same pathway in *Pseudomonas syringae* subsp. Savastanoi, and they observed a mutant with a production 5-fold higher than the wild-type strain. However, the effect on the root growth and branching of sorghum was negative. The authors confirmed that there is a limit to the concentration of IAA that is beneficial for root growth, and extremely large amounts can be detrimental.

Similar to the addition of tryptophan as a precursor of IAA, it is important to consider other relevant factors and parameters in the process. Prado et al. [195] carried out a factorial design $2^3$ in SSF, varying the percentage of tryptophan (0.5, 1.0, and 1.5) and particle size (in mm; 0.5, 1.0, and >1.0) and adding water (in mL; 5, 10, and 15). In this experiment, using the *A. flavipes* strain, it was observed that the amount of water and the use of tryptophan in the medium had a positive effect on the production of IAA in SSF. In addition, light and aeration influences have not been yet reported or described as parameters in the production of IAA in SSF.

## 6. Other Important Agricultural Bioinputs Obtained by SSF

In addition to the agricultural bioinputs obtained by SSF reported in this article, enzymes are also important products. Enzymes, along with inoculants and biological control agents, are commonly used in agriculture as a sustainable substitute for synthetic products. Agricultural enzymes help increase agricultural production, soil fertility, and food protection. They can also protect crops from various pests and diseases. The use of enzymes and inoculants increases the quality of commercial crops. Enzymes of agricultural importance, such as hydrolases, decompose plant residues and other organic matter that provide nutrients to plants and help in the initial stages of seed development, such as rooting and sprouting. Furthermore, the crop's resistance to water stress and nutrient assimilation are improved. Thus, enzymes are essential for sustainable agricultural productivity and soil management. The most important enzymes used in plant growth and soil fertility include phosphatases, dehydrogenases, and ureases. Other enzymes used in agricultural applications include cellulases, proteases, phytases, sulfatases, and amylases [200–202].

## 7. Future Prospects

The aforementioned bioinputs are emerging microbial-based technologies that are expected to come to the fore in the coming decades. Several fermentation strategies have been developed, different bioproducts have been exploited, and, yet, productivity and competitiveness are still concerns in relation to synthetic agrochemicals. Nevertheless, the agricultural systems rely on basic monocultures, and approaches comprising agroecological systems may be one of the alternatives to validate the use of bioinputs in a sustainable manner. Furthermore, one of the main factors affecting the efficiency of bioinputs is their persistence in soil. In fact, upon application, inoculants and biological control agents may find harsh conditions to survive, facing competition and suppression by the adapted microbial populations [203]. As such, current studies are attempting to overcome these issues via the development of smart bioformulations. Bioformulations are highly complex materials consisting of the main bioinputs (e.g., enzymes, spores, viable cells, and metabolites) entrapped in an organic matrix, such as natural polymers (e.g., cellulose, lignin, starch,

alginate, lectin, and others), which confers biodegradability and stabilization, and serves as a carrier/vehicle prolonging the shelf-life and persistence in soil under oscillatory environmental conditions. This will play a pivotal role in contributing to the commercialization of these bioinputs worldwide, increasing their affordability and broadening storage and application conditions.

## 8. Conclusions

As can be seen, in recent decades agricultural bioinputs have emerged as fundamental pieces in the search for more sustainable and efficient agricultural practices. The importance of these bioproducts spans economic, environmental, and geopolitical dimensions, reflecting a broad approach to addressing global food production challenges. These bioproducts are of fundamental importance for the development of the bioeconomy and circular economy, promoting the sustainable development of modern civilizations. As reported, inoculants, biological control agents, biosurfactants, and phytohormones play a crucial role in maximizing agricultural productivity by contributing to food security and increasing profitability for farmers. The use of agricultural bioinputs also reduces dependence on synthetic agrochemicals, minimizing adverse impacts on the biosphere and having a lower impact on biogeochemical cycles, contributing to more sustainable agricultural practices and mitigating climate change. Geopolitically, bioinputs are also important, as they give countries strategic advantages, reducing dependence on imports of chemical products and associated technologies. This strengthens the resilience of national agricultural systems, protecting countries from fluctuations in commodity prices and ensuring food sovereignty.

It should also be highlighted that SSF, as a production process for agricultural bioinputs, represents an innovative and effective approach in the production of agricultural bioinputs mainly associated with the context of biorefineries. By taking advantage of organic waste to promote the growth of beneficial microorganisms and the production of substances that influence plant growth, this process contributes to promoting soil health, increasing agricultural productivity, and reducing the environmental impact associated with conventional practices. FES stands out as a valuable component in the search for more sustainable and resilient agricultural systems.

Agricultural bioinputs are indispensable biotechnological products for a sustainable and less unequal future in food production, guaranteeing food security for different global civilizations.

**Author Contributions:** Writing—original draft, conceptualization, methodology, and writing—review and editing: T.M.R., P.R.F.M. and R.A.M.D.C. Validation, formal analysis, and data curation: F.G.B. Writing—original draft preparation: D.R.-R. Resources and supervision: S.S.d.S. All authors have read and agreed to the published version of the manuscript.

**Funding:** The authors gratefully acknowledge Coordenação de Aperfeiçoamento de Pessoal de Nível Superior–Brasil (CAPES) Finance Code 001; Fundação de Amparo à Pesquisa do Estado de São Paulo (FAPESP), Process number #16/10636-8 and #16/14852-7; and Conselho Nacional de Desenvolvimento Científico e Tecnológico (CNPq) with process number 141639/2023-7 for their financial support.

**Institutional Review Board Statement:** Not applicable.

**Informed Consent Statement:** Not applicable.

**Data Availability Statement:** The information contained in this review was surveyed from Google Scholar, Web of Science and Scopus database platforms.

**Acknowledgments:** The authors acknowledge the Laboratory of Bioprocesses and Sustainable Products.

**Conflicts of Interest:** The authors declare no conflicts of interest.

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
