# Peer review of "Agricultural Bioinputs Obtained by Solid-State Fermentation: From Production in Biorefineries to Sustainable Agriculture"

_sustainability, doi:10.3390/su16031076_

Round 1

Reviewer 1 Report

Comments and Suggestions for Authors

In the manuscript presented by Rocha et al., the authors presented sufficient information about the production of bioinputs by a solid-state fermentation system. The topic is attractive and I can attack many scientists interested in this field. I think this manuscript can be published with some minor following comments;

-Please add more case studies about the production of bioinputs by SSF.

-Please discuss the other fermentation methods and compare them by SFF, if it is available.  

Comments on the Quality of English Language

The English style is good enough. 

Author Response

The revision point by point can be found in the attached document 

Reviewer 2 Report

Comments and Suggestions for Authors

The title "Agricultural bioinputs obtained by solid state fermentation: from production in biorefineries to sustainable agriculture " seems clear and specific, providing a good overview of the paper's main focus.

Please rewrite the abstract from L18 to L21

please add some recent references supporting your objectives at the end of introduction. you can add https://doi.org/10.47278/journal.ijab/2023.071

There are several language errors like line 93 you used word more recently its inappropriate.

L96 to L98 please rewrite it

The language of the review reflects you used some AI software, please improve language carefully.

L105 With the popularization? what its means; same as L137 TO 139; L269 which is why it is routinely? same as line 270 you wrote According to the authors of the study?  L289 It should be noted?

Please add and include https://doi.org/10.47278/journal.ijab/2023.059 in Production of inoculants and biological control agents

There is much repetition of same statements

please add some recent references of 2022 and 2023

Please add a section of future prospects before conclusions

Comments on the Quality of English Language

Major revisions

Author Response

The comments were replied in the attached file

Reviewer 3 Report

Comments and Suggestions for Authors

Dear Authors,

I hope this message finds you well. I am writing to inform you that your manuscript, titled "Agricultural Bioinputs Obtained by Solid State Fermentation: From Production in Biorefineries to Sustainable Agriculture," and identified with Manuscript ID sustainability-2808220, has been accepted for publication in the journal "Sustainability" (ISSN 2071-1050). Your submission has been categorized under the special issue "Technologies for a Sustainable Future: Bioeconomy, Biorefineries, Biobased Products, and Energy" in the section of "Resources and Sustainable Utilization."

I commend the quality and significance of your work, and I am pleased to confirm its acceptance. However, I recommend some minor revisions, primarily focused on grammatical improvements. Additionally, I kindly request further clarification for tables three and four, as their content lacks explicit detail, making it challenging for readers to comprehend their significance within the context of the study.

Addressing these aspects in your revision will undoubtedly enhance the overall quality of the article, thereby contributing significantly to the advancement of processes in modern biorefineries and playing a pivotal role in the mitigation of environmental impacts.

I appreciate your attention to these suggestions and look forward to receiving the revised version of your manuscript.

Best regards,

Comments on the Quality of English Language

Dear Authors,

I hope this message finds you well. I am writing to inform you that your manuscript, titled "Agricultural Bioinputs Obtained by Solid State Fermentation: From Production in Biorefineries to Sustainable Agriculture," and identified with Manuscript ID sustainability-2808220, has been accepted for publication in the journal "Sustainability" (ISSN 2071-1050). Your submission has been categorized under the special issue "Technologies for a Sustainable Future: Bioeconomy, Biorefineries, Biobased Products, and Energy" in the section of "Resources and Sustainable Utilization."

I commend the quality and significance of your work, and I am pleased to confirm its acceptance. However, I recommend some minor revisions, primarily focused on grammatical improvements. Additionally, I kindly request further clarification for tables three and four, as their content lacks explicit detail, making it challenging for readers to comprehend their significance within the context of the study.

Addressing these aspects in your revision will undoubtedly enhance the overall quality of the article, thereby contributing significantly to the advancement of processes in modern biorefineries and playing a pivotal role in the mitigation of environmental impacts.

I appreciate your attention to these suggestions and look forward to receiving the revised version of your manuscript.

Best regards,

Author Response

The comments are presented in the attached file

Reviewer 4 Report

Comments and Suggestions for Authors

The manuscript is well prepared and the presented results are discussed properly. The topic of the study seems to attract the attention of the Sustainability readers. The 19% similarity is acceptable. I have not found any major issue in the manuscript and mostly it has been developed scientifically. Therefore, I suggest acceptance of the manuscript after brief English proofreading.

Comments on the Quality of English Language

Minor editing of the English language required

Author Response

The comments can be found in the attached document

Round 2

Reviewer 2 Report

Comments and Suggestions for Authors

Author did good job